# A *k*-mismatch string matching for generalized edit distance using diagonal skipping method

**HyunJin Kim** [ID]*

School of Electronics and Electrical Engineering, Dankook University, Yongin, Gyeonggi, Republic of Korea

* hyunjin2.kim@gmail.com

## Abstract

This paper proposes an approximate string matching with *k*-mismatches when calculating the generalized edit distance. When the edit distance is generalized, more sophisticated string matching can be provided. However, the execution time increases because of the bundle of complex computations for calculating complicated edit distances. The computational costs for finding which steps or edit distances are over *k*-mismatches cannot be significant in the generalized edit distance metric. Therefore, we can reduce the execution time by determining steps over *k*-mismatches and then skipping them. The diagonal step calculations using the pruning register skips unnecessary distance calculations over *k*-mismatches. The overhead of control statements and reordered memory accesses can be amortized by skipping multiple steps. Even though the proposed skipping method requires additional overhead, the proposed scheme's practical embodiments show that the execution time of string matching is reduced significantly when *k* is small.

**Data Availability Statement:** All code files are available from the GitHub database (https://github. com/analog75/ED).

**Funding:** The author received no specific funding for this work.

## Introduction

In the field of computer science, information retrieval is a fundamental problem. Notably, string matching is essential to digital information retrieval. String matching searches the sequence of characters or pattern to determine whether the pattern matches with an input sequence or not. In exact string matching, when a pattern is the same as an input sequence, it determines that the pattern is matched with the input sequence. On the other hand, approximate string matching evaluates the similarity between the input sequence and pattern based on its metric. With sophisticated data analysis and various applications, the approximate string matching can get more attention in the big-data era [1–6].

The similarity between two strings can be quantitated by the minimum number of basic operations that makes an input sequence equal to the target pattern. Traditionally, approximate string matching assumes that the insertion, deletion, replacement, and transposition of characters in a string make the difference [1, 7]. They are used as basic operators to calculate the distance between the input sequence and target pattern. In the Levenshtein distance calculation, string matching can be simplified because each basic operator has the unified cost of one. When estimating the distance between two strings. the Hamming distance [8] calculation counts '1' bits after applying bitwise exclusive-OR.

**Competing interests:** The authors have declared that no competing interests exist.

On the other hand, the semantics or relationship between subsequences make approximate string matching sophisticated. For example, a human can feel that pattern "catch" is more similar to input sequence "cotch" than to input sequence "ctch" although the Levenshtein distance from pattern "catch" is one for both two input sequences, respectively. Therefore, a more complicated edit distance metric can be adopted, categorized into the *generalized edit distance* [9, 10] or the *normalized edit distance* [11, 12]. However, when complex functions generalize the edit distance, significant computational resources are required. When calculating the edit distance between input sequence and pattern, the edit distance between each input subsequence and subpattern is needed, which is called *step*. Moreover, in the traditional sequential dynamic programming [13], all steps should be calculated in order, which is a very time-consuming job due to the data dependency in calculating steps. Several mathematical approaches can show better computational complexity [14, 15]. However, the overhead of control statements and reordered memory accesses is not considered for practical applications. The parallel string matching methods have been researched using the parallelism equipped in GPU (Graphics Processing Unit) [7, 16–24] and FPGA (Field Programmable Gate Array) [25–29], where the parallel programming requires multiple computational resources. However, sequential string matching based on a processing unit is still an attractive and fundamental topic in many practical applications. Our study reduces the execution time of the sequential approximate string matching when performed by a processing unit.

Naively, in a step calculation, if its data-dependent previous steps have over *k*-mismatches, the evaluation of operators with these previous steps over *k* can be skipped. However, in the Levenshtein distance metric, the overhead ratio for finding whether data-dependent steps are over *k*-mismatches is relatively high. In previous theoretical approaches [14, 15], this overhead is not considered, so their implementations cannot have better performance than that of the dynamic programming-based method [13] in the Levenshtein distance metric. With the generalized edit distance, more sophisticated string matching can be confirmed. The execution time increases because of the bundle of complex computations for performing complex edit calculations. Therefore, if the step calculation that is expected to be over *k* can be skipped, the total execution time can be significantly reduced, which motivates our research.

This paper proposes an approximate string matching with *k*-mismatches for the edit distance metric. Our research is motivated that when previous steps are over *k*, the information can be used to skip unnecessary step calculations. This paper focuses on the practical embodiment of our method and its evaluation. Without finding which data-dependent previous steps are over *k*, the diagonal step calculations using the pruning register can skip unnecessary step calculations over *k*-mismatches. Each bit in the pruning register contains the information of step calculations to be skipped. Even though there is an additional overhead of control statements and reordered memory accesses, skipping multiple steps at a time can reduce execution time significantly. For realistic experiments, generalized edit distance metrics are assumed based on the similarity in shapes and keyboard character positions. The proposed string matching and other dynamic programming methods are coded and then evaluated using the generalized edit distance metrics. Despite additional overhead in the diagonal step calculations and pruning register accesses, experiments show that the proposed skipping method can reduce the execution time of approximate string matching when *k* is small.

## Preliminaries

### Edit distance in approximate string matching

In string matching, an input sequence is compared with the pattern, and then the difference between the input sequence and pattern is reported. Unlike exact string matching, the

similarity is quantified in the approximate string matching. The distance between the input sequence and pattern refers to the calculation result based on the distance metric adopted in string matching.

In [1], for strings $X_i = x_1, x_2, \ldots, x_{i-1}, x_i$ and $Y_j = y_1, y_2 \ldots, y_{j-1}, y_j$ where characters $x_a, y_b \in \mathbb{C}$ for $1 \leq a \leq i$ and $1 \leq b \leq j$, the distance between $X_i$ and $Y_j$ denoted as $D(X_i, Y_j)$ is the minimum number of edit operations to make $X_i$ and $Y_i$ the same. The distance $D(X_i, Y_j)$ should satisfy:

- $D(X_i, Y_j) = 0$ if and only if $X_i = Y_j$

- $D(X_i, Y_j) > 0$ when $X_i \neq Y_j$

- $D(X_i, Y_j) = D(Y_j, X_i)$.

Besides, for a given string $Z_k = z_1, z_2, \ldots, z_{k-1}, z_k$ where $z_c \in \mathbb{C}$ for $1 \leq c \leq k$, the edit distance satisfies the condition of $D(X_i, Z_k) \leq D(X_i, Y_j) + D(Y_j, Z_k)$ called triangle inequality.

Significantly, the Levenshtein distance [30] is the most popular edit distance metric in string matching, so the edit distance has been interchangeably used with the Levenshtein distance sometimes. However, because the edit distance can include several meanings of other metrics different from the Levenshtein distance metric, this paper denotes that the Levenshtein distance metric adopts simple operators with the cost of one.

We define input subsequence $X_\alpha$ of input sequence $X_i$ and subpattern $Y_\beta$ of pattern $Y_j$ for $1 \leq \alpha \leq i$ and $1 \leq \beta \leq j$, as follows:

**Definition 1** *For strings $X_\alpha = x_1, x_2, \ldots, x_{\alpha-1}, x_\alpha$, and $Y_\beta = y_1, y_2, \ldots, y_{\beta-1}, y_\beta$, when subscripts $\alpha \leq i$ and $\beta \leq j$, $X_\alpha$ and $Y_\beta$ are the input subsequence and subpattern of input sequence $X_i$ and pattern $Y_j$, respectively.*

For input subsequence $X_\alpha$ and subpattern $Y_\beta$, when the initial edit distance $D(X_0, Y_0)$ is 0, the minimum edit distance $D(X_\alpha, Y_\beta)$ is formulated as follows:

$$D(X_\alpha, Y_\beta) = min \begin{cases} D(X_{\alpha-1}, Y_{\beta-1}) + substitution(x_\alpha, y_\beta) \\ D(X_{\alpha-1}, Y_\beta) + deletion(x_\alpha) \\ D(X_\alpha, Y_{\beta-1}) + insertion(y_\beta). \end{cases} \tag{1}$$

Black arrows 1, 2, and 3 in Fig 1(a) mean substitution, deletion, and insertion operators, which correspond to cost functions $substitution(x_\alpha, y_\beta)$, $deletion(x_\alpha)$, and $insertion(y_\beta)$ in Eq (1), respectively. The function $substitution(x_\alpha, y_\beta)$ means the cost of substituting $x_\alpha$ of $X_\alpha$ into $y_\beta$ of $Y_\beta$. The function $deletion(x_\alpha)$ provides the cost of deleting $x_\alpha$ from $X$. On the other hand, the function $insertion(y_\beta)$ means the cost of inserting $y_\beta$ to the end of $Y_{\beta-1}$.

For example, let's assume that $X_3 = $ "bat" and $Y_3 = $ "bad" with $\alpha = \beta = 3$. In this case, by substituting $x_3$('*t*') with $y_3$('*d*') in $X_3$ in $substitution(x_3, y_3)$, the converted $X_3$ can be the same as $Y_3$, which adds the cost of $substitution$('*t*', '*d*') to $D$("*ba*", "*ba*"). When $D$("*ba*", "*bad*") is given, character $x_3$('*t*') is removed from "bat", and the given $D$("*ba*", "*bad*") is required to convert "ba" to "bad". Therefore, the cost of $deletion$('*t*') is added to calculate $D$("*bat*", "*bad*"). When $D$("*bat*", "*ba*") is given, after attaching $y_3 = $ '*d*' to "ba", $D$("*bat*", "*bad*") can be calculated, which means that the cost of $insertion$('*d*') should be added.

|     |     |   | pattern |   |   |   |   |
|-----|-----|---|---|---|---|---|---|
|     |     |   | c | a | t | c | h |
|     |     | 0 | 1 | 2 | 3 | 4 | 5 |
|     | c   | 1 | 0 | 1 | 2 | 3 | 4 |
|     | c   | 2 | 1 | 1 | 2 | 2 | 3 |
| input sequence | a | 3 | 2 | 1 | 2 | 3 | 3 |
|     | t   | 4 | 3 | 2 | 1 | 2 | 3 |
|     | e   | 5 | 4 | 3 | 2 | 2 | 3 |
|     | s   | 6 | 5 | 4 | 3 | 3 | 3 |
|     | e   | 7 | 6 | 5 | 4 | 4 | 4 |

(a)  (b)

**Fig 1. An example of the Levenshtein distance matrix for input sequence "ccatese" and pattern "catch" (a) operators (b) Levenshtein distance matrix.**

Therefore, the minimum edit distance for $X_3$ = "bat" and $Y_3$ = "bad" formulated as $D($"*bat*", "*bad*"$)$ is calculated on Eq (1) as:

$$D(bat, \ bad) = min \begin{cases} D(ba, \ ba) + substitution(t, \ d) \\ D(ba, \ bad) + deletion(t) \\ D(bat, \ ba) + insertion(d). \end{cases} \quad (2)$$

The Levenshtein distance metric simplifies the cost of each operator into 1 or 0, which makes the Levenshtein distance calculation very simple. Fig 1(b) illustrates an example of the Levenshtein distance matrix. In Fig 1(b), input subsequence "ca" can be the same as subpattern "cat" after attaching "t" to the end of input subsequence "ca". When using the insertion operator, an input subsequence can be equal to the subpattern. Substitution and deletion operators are also applied to the input sequence to match with the pattern. In Fig 1(b), the rightmost bottom cell is numbered as 4, which is the final Levenshtein distance between input sequence "ccatese" and pattern "catch".

We denote the edit distance between input subsequence and subpattern as *step*. In a *traversal*, the steps included in the traversal are calculated in order. The traversal method determines the order of calculating steps.

## Generalized edit distance

Unlike the Levenshtein distance, the generalized edit distance adopts more sophisticated cost functions in Eq (1). Depending on the operator type, Cost functions can output other values different from 1 or 0. If the cost of one deletion operation increases twice as much as that of one substitution operation, it follows as:

$$substitution(x_\alpha, y_\beta) = 2 \cdot deletion(x_\alpha). \tag{3}$$

For example, the similarity in character shapes can be used in another generalized distance metric. Because 'h' is similar to that of 'b' in shape, a human can feel that input sequence "catcb" is more similar to pattern "catch" than input sequence "catco". In this case, the similarity can be estimated by the substitution operator in the generalized edit distance metric. In another example, the misspelling can happen depending on the character positions in a keyboard. In the US computer keyboard, 'q' has a high possibility of being mistyped as 'w' because the key of 'w' is located next to that of 'q'. However, the shape of 'p' is totally different from that of 'w', so that other functions are needed to quantify the difference between key positions. Besides, the generalized edit distance can consider the pattern length. Intuitively, we feel that the difference between "ca" and "cat" is expected to be greater than that between "catasrophe" and "catastrophe" even though the difference from both cases is caused by one deleted character 't'. Therefore, the costs from the insertion and deletion operations can be inversely proportional to the pattern length. In this case, condition $D(X_\alpha, Y_\beta) = D(Y_\alpha, X_\beta)$ cannot be met when the costs for insertion and deletion operations are different from each other. In conclusion, the generalized edit distance metric requires more complicated operations. Besides, these generalized edit distance metrics can adopt fractions to represent the distance.

Fig 2 shows an example of the edit distance matrix using the generalized edit distance metric based on the similarity in shape and pattern length. We assume that the costs of deletion and insertion operators are fixed as 0.76. For the substitution operation, different values are added depending on character shapes. In Fig 2, because characters 'c' and 'e' seem to be similar in shape, *substitution*('c', 'e') = 0.42. On the other hand, for the cases with characters 'a' and 'h', *substitution*('a', 'h') = 1.20. Unlike Fig 1, each step's distance has a fraction in Fig 2, so that the generalized edit distance calculation needs fractional operations. As these operators need complex computations, the evaluation of each operator requires an additional computational overhead.

## *k*-mismatch string matching

A *k*-mismatch approximate string matching is defined as:

**Definition 2** *In k-mismatch string matching, for input sequence $X_i$ and pattern $Y_j$, when $D(X_i, Y_j) \leq k$, $X_i$ matches $Y_j$.*

Term *k* denotes the threshold for determining whether $X_i$ is matched with $Y_j$ or not.

Because the cost of any operation is a positive value, Eq (1) can be modified for *k*-mismatch string matching with input subsequence $X_\alpha$ and subpattern $Y_\beta$ as:

$$D(X_\alpha, Y_\beta)_k = min \begin{cases} D(X_{\alpha-1}, Y_{\beta-1})_k + substitution(x_\alpha, y_\beta) \text{ when } D(X_{\alpha-1}, Y_{\beta-1})_k \leq k \\ \\ D(X_{\alpha-1}, Y_\beta)_k + deletion(x_\alpha) \text{ when } D(X_{\alpha-1}, Y_\beta)_k \leq k \\ \\ D(X_\alpha, Y_{\beta-1})_k + insertion(y_\beta) \text{ when } D(X_\alpha, Y_{\beta-1})_k \leq k. \end{cases} \tag{4}$$

In Eq (4), when the edit distance of a data-dependent previous step ($D(X_{\alpha-1}, Y_{\beta-1})_k$, $D(X_{\alpha-1}, Y_\beta)_k$, and $D(X_\alpha, Y_{\beta-1})_k$) is over *k*, there is no need to evaluate its operation for calculating *D*

**pattern**

|  | | c | a | t | c | h |
|---|---|---|---|---|---|---|
|  |  | 0.00 | 0.76 | 1.52 | 2.28 | 3.04 | 3.80 |
|  | c | 0.76 | 0.00 | 0.76 | 1.52 | 2.28 | 3.04 |
|  | c | 1.52 | 0.76 | 0.75 | 1.51 | 1.52 | 2.28 |
| input sequence | a | 2.28 | 1.52 | 0.76 | 1.52 | 2.26 | 2.71 |
|  | t | 3.04 | 2.28 | 1.52 | 0.76 | 1.52 | 2.28 |
|  | e | 3.80 | 3.04 | 2.28 | 1.52 | 1.18 | 1.94 |
|  | s | 4.56 | 3.80 | 3.04 | 2.28 | 1.94 | 2.70 |
|  | e | 5.32 | 4.56 | 3.80 | 3.04 | 2.70 | 3.46 |

**Fig 2. Example of generalized edit distance matrix for input sequence "ccatese" and pattern "catch".**

$(X_\alpha, Y_\beta)_k$. However, an additional overhead is required to find whether its data-dependent previous steps are over $k$ or not.

## Proposed diagonal string matching using pruning register

### Motivations

From Eq (4), when the edit distance of data-dependent previous steps ($D(X_{\alpha-1}, Y_{\beta-1})$, $D(X_{\alpha-1}, Y_\beta)$, $D(X_\alpha, Y_{\beta-1})$) over $k$-mismatches is pre-known, we determine whether its related operation is needed or not. Our motivation starts from the fact that unnecessary step calculations over $k$-mismatches can be skipped depending on data-dependent previous steps. In the existing dynamic programming-based method, the distance matrix is filled by calculating edit distances between input subsequences and subpatterns, so that the data-dependent previous steps are accessed in the edit distance matrix.

Fig 3 is the conceptual figure that illustrates overhead ratios of conditional statements for finding data-dependent previous steps according to the computational overhead for performing operations (substitution, insertion, and deletion). In a simple edit distance metric such as

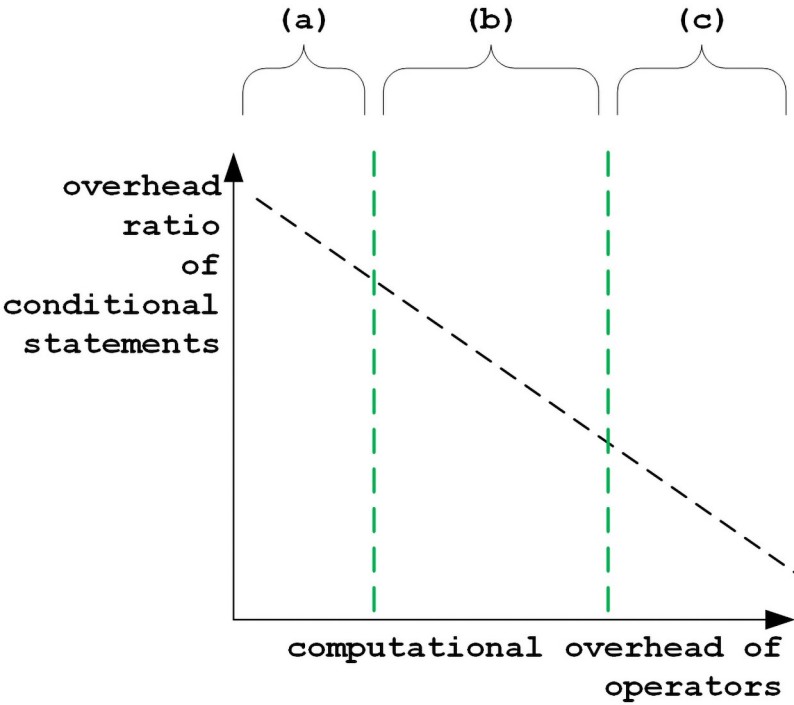

**Fig 3. Overhead ratios of conditional statements according to computational overhead for performing operations.**

the Levenshtein distance, each operation only compares characters and makes binary output, so that several operations are performed in the pipelining [31]. If there is any conditional jump, these predicted instructions of many operations can be cancelled, which degrades the performance. In other words, it is expected that the conditional jump for skipping evaluations cannot reduce the total execution time. The overhead ratio of conditional statements for finding whether a data-dependent previous step is over *k* is too high. Therefore, in a simple edit distance metric, there could be no benefits by skipping evaluations in range (a) of Fig 3.

On the other hand, when the edit distance metric requires more computational resource to evaluate complicated operators, the skipping method can be useful. As the computational resources for performing each operator increase, the overhead ratio of conditional statements becomes very small. In range (c), it can be better to skip each operator evaluation when finding its data-dependent previous steps over *k*. In range (b), the overhead of conditional statements and operator evaluations is not negligible. If the remaining iterations can be skipped in the loop for calculating each step, many operator evaluations can be reduced. The implementation of dynamic programming [13] using a nested loop cannot provide such functions.

Therefore, we propose a new string matching method for skipping the remaining iterations for the distance metric. In the following, the problem definition is discussed in detail, and the proposed diagonal skipping method is explained.

## Problem definition

Fig 4 shows examples of calculating steps and their traversals considering data dependency between steps, in which Fig 4(a) shows simple vertical traversals. In vertical traversal, steps on the next column depend on those on the previous column for substitution and insertion operators. Therefore, after calculating steps on a column, steps neighbouring on the right column

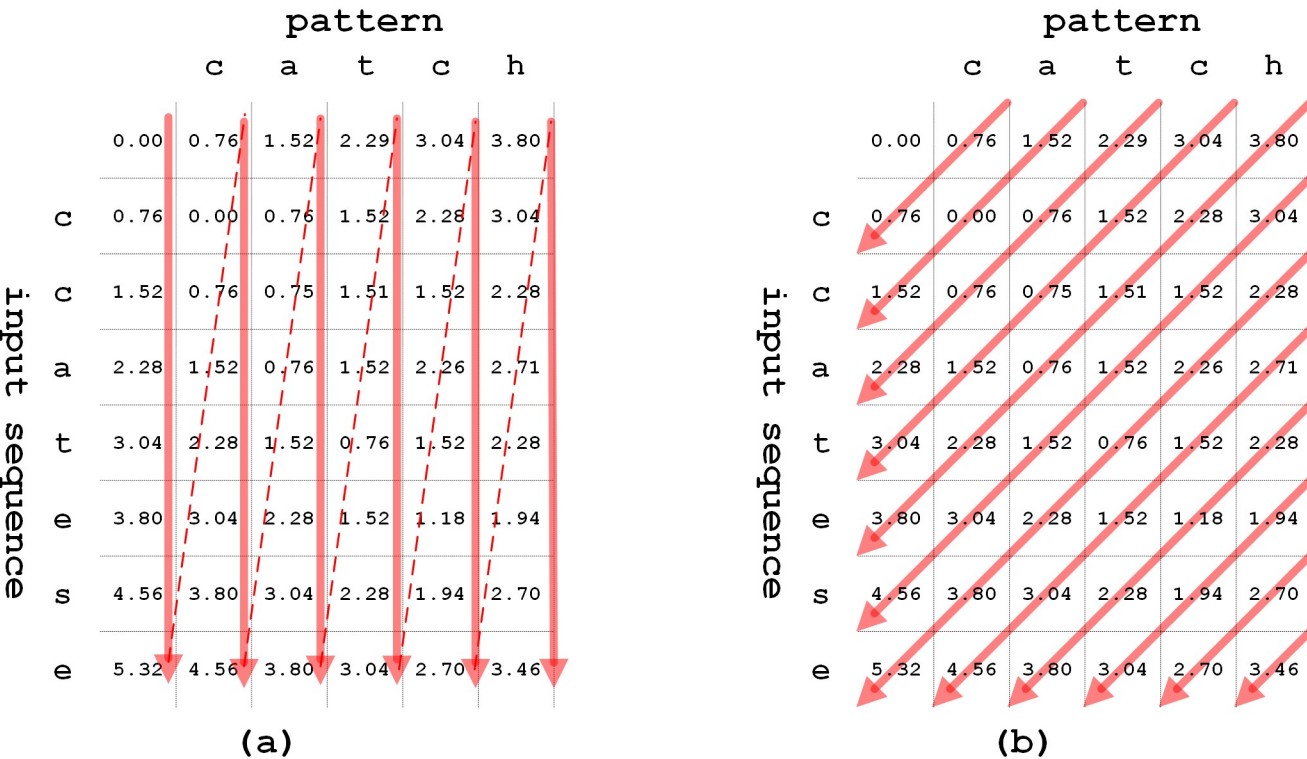

**Fig 4. Step calculations in edit distance matrix; (a) vertical traversals: (b) diagonal traversals.**

can be calculated. Besides, for the deletion operator, the traversal should proceed from top to bottom. This dynamic programming considers data dependency between its neighbouring steps that exists from Eq (1) [13]. After calculating steps in a traversal, the vertical traversal is performed on the right column. Therefore, when *n* and *m* are denoted as the input sequence and pattern lengths, the computational complexity can be $O(mn)$.

Several approximate string matching algorithms have been studied to reduce the dynamic programming's computational complexity [14, 15]. In general, several previous works about *k*-mismatch string matching enhance the throughput of string matching for the long input string such as network traffic data [2] and DNA sequences [4, 5]. Therefore, multiple occurrences of the pattern are searched in the long input string, where string matching with input subsequences can be considered. For example, when input sequence and pattern are "baseball player" and "catastrophe", the Levenshtein distance between subsequence "base" and pattern "catastrophe" is 10. Considering the distance of 10, if *k* < 10, input subsequence "baseball" cannot be matched. Then, another string matching with another subsequence "player" begins. In this case, the calculation of the edit distance matrix cannot be avoidable. This paper proposes a new method that reduces the execution time of obtaining the edit distance matrix for *k*-mismatches.

## Diagonal traversal and skipping method

Our method adopts the diagonal traversal to skip unnecessary step calculations over *k*-mismatches. Unlike the vertical traversal performed on each column, the diagonal traversal calculates steps across columns. Even though the work in [14] proposes the diagonal evaluation

based on reordered data structure, the step calculations are not skipped for *k*-mismatch string matching.

In Fig 4(b), diagonal traversals are illustrated, where an arrow illustrates the order of steps calculated in each traversal. Fig 4(b) describes that the upper right step is calculated before the lower left step in a diagonal traversal. It is denoted that *t* is the index of a traversal, and the traversals indicated by arrows *traversal*($t - 2$), *traversal*($t - 1$), and *traversal*($t$) are performed in order. The calculations of steps on a diagonal traversal *traversal*($t$) do not have data dependency with each other. Each step calculation of *traversal*($t$) has data dependency with three steps of *traversal*($t - 1$) and *traversal*($t - 2$). For substitution operation, a step can be calculated after obtaining the step in *traversal*($t - 2$). For insertion and deletion operations, two steps can be calculated depending on steps of *traversal*($t - 1$). These calculations require the values of two steps for the substitution and insertion operations on the left column and one step for deletion operation on the same column.

When the previous diagonal traversals *traversal*($t - 2$) and *traversal*($t - 1$) finish the calculations of all data-dependent previous steps, *traversal*($t$) can use the calculation results to skip unnecessary operator evaluations. Our proposed method adopts so-called *pruning register* to avoid multiple iterations in a loop without accessing each element in the edit distance matrix. Each pruning bit in the pruning register is assigned into a column of the edit distance matrix. When the pruning bit for its column is set as '1', there is no need to calculate all steps in the column. In this case, the steps to be calculated have distances over *k*. The pseudocode of the proposed string matching is as follows:

**Algorithm 1** Diagonal Skipping

```
 1: procedure DIAGONAL_SKIPPING (input sequence, pattern)
 2:    Initialization(D[0, ...,i][0], D[0][1, ...,j], pruning_reg)
 3:    for each traversal do
 4:       for each step(α, β) ∈ traversal do
 5:          if pruning_reg(β) == 1 then break;
 6:          end if
 7:          if pruning_reg(β - 1) == 0 then
 8:             D[α][β] = cost_min((D[α - 1][β], D[α][β - 1], D[α - 1][β - 1]))
 9:          else
10:             D[α][β] = cost_min((D[α - 1][β], D[α - 1][β - 1]))
11:          end if
12:          if D[α][β] > k and pruning_reg(β - 1) == 1 then
13:             pruning_reg(β) = 1;
14:          end if
15:       end for
16:    end for
17:    return D
18: end procedure
```

In the pseudocode, the procedure **Diagonal_Skipping** has two arguments: input sequence and pattern. Terms *i*, *j*, and *k* denote the input sequence length, pattern length, and mismatch threshold *k*, respectively. Firstly, several elements in two-dimensional $(i + 1) \times (j + 1)$ array *D* and pruning register *pruning_reg* are initialized. In this initialization, $D[0, \ldots, i][0]$ is initialized using only deletion operators. On the other hand, $D[0][1, \ldots, j]$ is initialized using only insertion operators. These steps can be simply calculated without considering *min*() function in Eq (1). In the pruning register, the bit indicating the leftmost column (the 0-th column) is set as '1', and other bits are set as '0'.

Then, each step in a *traversal* is calculated in order. The direction of the arrow is from right top to left bottom, as shown in Fig 4. As shown in *Preliminaries* section, $X_\alpha$ and $Y_\beta$ mean a subsequence of input sequence $X_i$ and a subpattern of pattern $Y_j$ for $1 \le \alpha \le i$ and $1 \le \beta \le j$. An

element $D[\alpha][\beta]$ in the two-dimensional array contains edit distance $D(X_\alpha, Y_\beta)$. For each step of $D[\alpha][\beta]$ for distance $D(X_\alpha, Y_\beta)$, if the pruning bit of the $\beta$-th column is '1', the next steps calculated in a traversal can be over $k$. Therefore, the **break** statement means that this procedure skips other iterations that calculate steps of the traversal; otherwise, each step in the traversal is calculated. When the pruning bit of ($\beta$–1)-th column is '0', data-dependent previous steps are accessed, and function $cost_{min}$ calculates the minimum distance. Except for the calculation $D[s][1]$, $1 \le s \le i$, when the pruning bit of ($\beta$–1)-th column is '1', the value stored in $D[\alpha][\beta - 1]$ is over $k$, so only $D[\alpha - 1][\beta]$ and $D[\alpha - 1][\beta - 1]$ are accessed to calculate $D[\alpha][\beta]$. When calculating $D[s][1]$, $1 \le s \le i$, all operators are considered because the pruning bit of the leftmost column is initialized as '1'. If the edit distance in $D[\alpha][\beta]$ is over $k$ and the pruning bit of the ($\beta - 1$)-th column is '1', the pruning bit of the $\beta$-th column becomes '1'. The number of diagonal traversals is proportional to the pattern length $j$. Therefore, the computational complexity can be $O(jk)$, which means that the computations can be mainly limited by $j$ and $k$.

Fig 5 illustrates the string matching operation with input sequence "ccatese" and pattern "catch". Firstly, the bit for the leftmost column in the pruning register is initialized as '1', as shown in Fig 5(a). Additionally, $D[0, \ldots, i][0]$ and $D[0][1, \ldots, j]$ are initialized. As diagonal traversals proceed on the fourth arrow, the step over $k = 2$ is reached on the second left column. Also, $pruning\_reg(1)$ is updated as '1', as shown in Fig 5(c). On the fifth arrow, because the

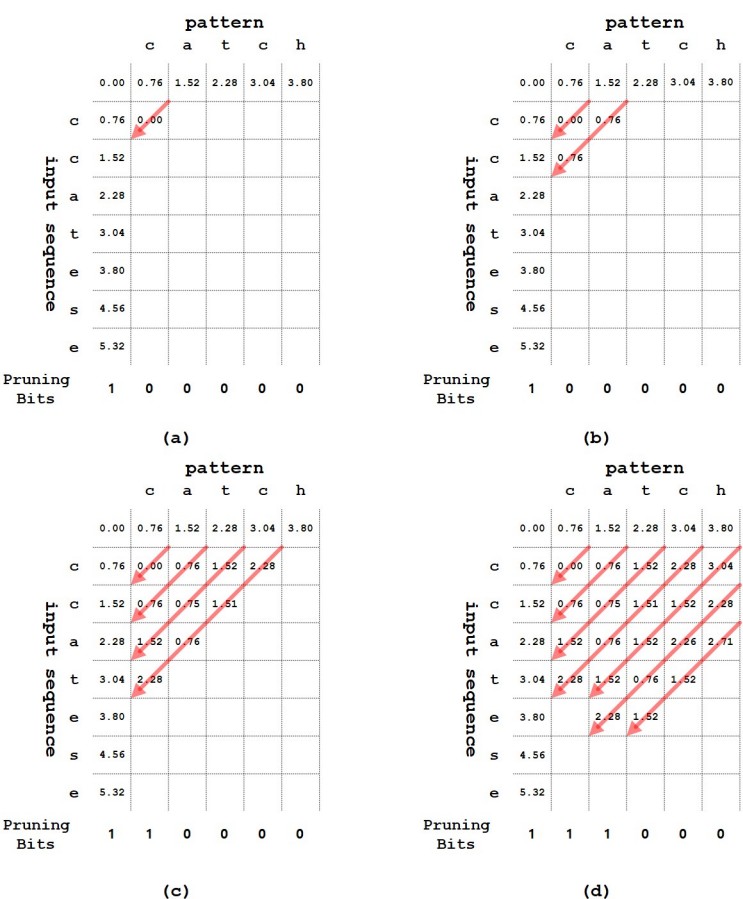

**Fig 5. An example of proposed method using pruning register.**

pruning bit of *pruning_reg*(1) is '1', function *cost*(*D*[3][2], *D*[3][1]) is performed to calculate *D*([4][2]), where the insertion operation is skipped. Fig 5(d) shows intermediate progress based on Algorithm 1. The proposed diagonal skipping method can determine the steps to be skipped just by accessing the pruning register instead of using all data-dependent previous steps. Also, the proposed method skips multiple-step calculations at a time, which reduces the execution time.

In the proposed method, the number of traversals on arrows is proportional to the pattern length *j*, where several step calculations over *k* are not skipped in the proposed method. If all neighbouring data-dependent steps on the same column over *k* are checked before evaluating operators, the unnecessary operator evaluations can be skipped, which can make the complexity $O(min(j, k))$. However, unlike the proposed diagonal skipping method using only one pruning register, complicated conditional statements and additional memory accesses are required. This method can be valid when the computational overhead of operator evaluations is significant, which is described in the range (c) of Fig 3.

## Experimental results and analysis

Based on realistic environments, we show the experimental results depending on different edit distance metrics. Firstly, when Levenshtein distance is calculated, it is expected that the skipping method is not effective due to the overhead of conditional statements. Then, when adopting the generalized edit distance metrics considering the visual similarity in shapes or keyboard character positions, the proposed skipping method can show better performance than the dynamic programming for small *k*-mismatches and the method using the reordered data structure. Besides, the overhead of conditional statements for finding data-dependent previous steps is discussed.

### Experimental environments

In experiments, the proposed method was coded and complied by C language and GCC 5.4.0, respectively. For apple-to-apple comparisons, we implemented the dynamic programming and the skipping method that found neighbouring data-dependent previous steps to skip each step calculation over *k*. These implemented codes have been uploaded in [32], where the execution times of the proposed method and other counterparts were measured. The tests were performed on a single core of Intel Xeon CPU E5-2630 v3 @ 2.40GHz machine with 16 Gigabyte main memory and Ubuntu 16.04 operating system. The experiments randomly selected 100,000 pairs of the input sequence and pattern from the English dictionary with 370,099 words [33], where the average and standard deviation of the input sequence and pattern lengths were 9.4 and 2.90, respectively.

We evaluated the proposed method based on three different distance metrics. Firstly, we calculated the Levenshtein distances to know the benefits of the proposed method in the simple edit distance metric. Secondly, for the evaluation using a highly computational edit distance metric, the similarity in shapes between two alphabet characters was quantified in a two-dimensional array *D* considering [34]. This array was used to calculate the cost in the substitution operator. For example, *substitution*('a', 'b') = 1/2.13 * *C* and *substitution*('o', 'e') = 1/4.13 * *C*, where *C* was the scaling factor for normalizing the substitution cost. In this example, the cost of *substitution*('a', 'b') can be 1.94 times the cost of *substitution*('o', 'e'). For insertion and deletion operators, this experiment assumed a weighted cost depending on pattern length *j*,

**Table 1. Features of evaluated edit distance metrics.**

| Metric | Role | Costs |
|---|---|---|
| Levenshtein | simple low-cost edit distance metric | low |
| Shape | normalized weighted edit distance metric | medium |
| Keyboard | complex weighted edit distance metric | high |

where we developed an exponential cost function using the average word length as:

$$deletion(x_\alpha), insertion(y_\beta) = \frac{exp\left(\dfrac{average_length}{j}\right)}{exp(1)}. \tag{5}$$

In Eq (5), costs were normalized by *exp*(1). As *j* increased, the costs of insertion or deletion operations decreased exponentially, so that different weights were assigned depending on *j*.

Finally, our experiments adopted a more complicated distance metric that considered character positions in a keyboard. In this metric, the Euclidean distance between characters was calculated to obtain each substitution operation's cost. The position of each character was stored in an array, which was used to calculate the Euclidean distance between characters. Based on the typo distance in [35], the function for calculating the cost of each substitution operation was implemented. Unlike typo distance [35] without commutative property, our edit distance metric had the same cost for the deletion and insertion operations to meet the edit distance's characteristic. The features of edit distance metrics above are summarized in Table 1.

## Experimental analysis

Fig 6 shows the summary of average execution times by sweeping *k* when using the Levenshtein distance metric. When the diagonal traversal did not adopt *k*-mismatches, the average execution time was longer than that of the dynamic programming using the vertical traversal because of the overhead from conditional statements and reordered memory accesses. In these experiments, the execution time increased with *k*. When *k* > 4, the execution time was over that of the vertical traversal, which means the proposed method did not have any benefits over the simple vertical traversal for large *k*. Besides, Fig 6 shows that the diagonal traversal without considering *k*-mismatches required the additional overhead of conditional statements and reordered data accesses compared with the vertical traversal. Therefore, for the Levenshtein distance metric, when *k* was small, we concluded that the proposed method can help reduce the execution time. Significantly, compared with the vertical and diagonal traversals, the execution times were decreased by 44.3% and 52.3% with *k* = 1.

For the generalized edit distance using similarity in shapes, Fig 7 illustrates the average execution times by sweeping *k*. Like the case using the Levenshtein distance metric, the execution time increased with *k*. When *k* < 5, the average execution times of the proposed diagonal skipping method were shorter than that of the vertical traversal, which means that many step calculations can be skipped for small *k* in this generalized edit distance metric. The diagonal traversal only increased the average execution time by 11.5% over the vertical traversal. Notably, when *k* = 1, the execution times were reduced by 55.7% and 60.3% over the vertical and diagonal traversals. Compared with the evaluation using the Levenshtein distance metric, it

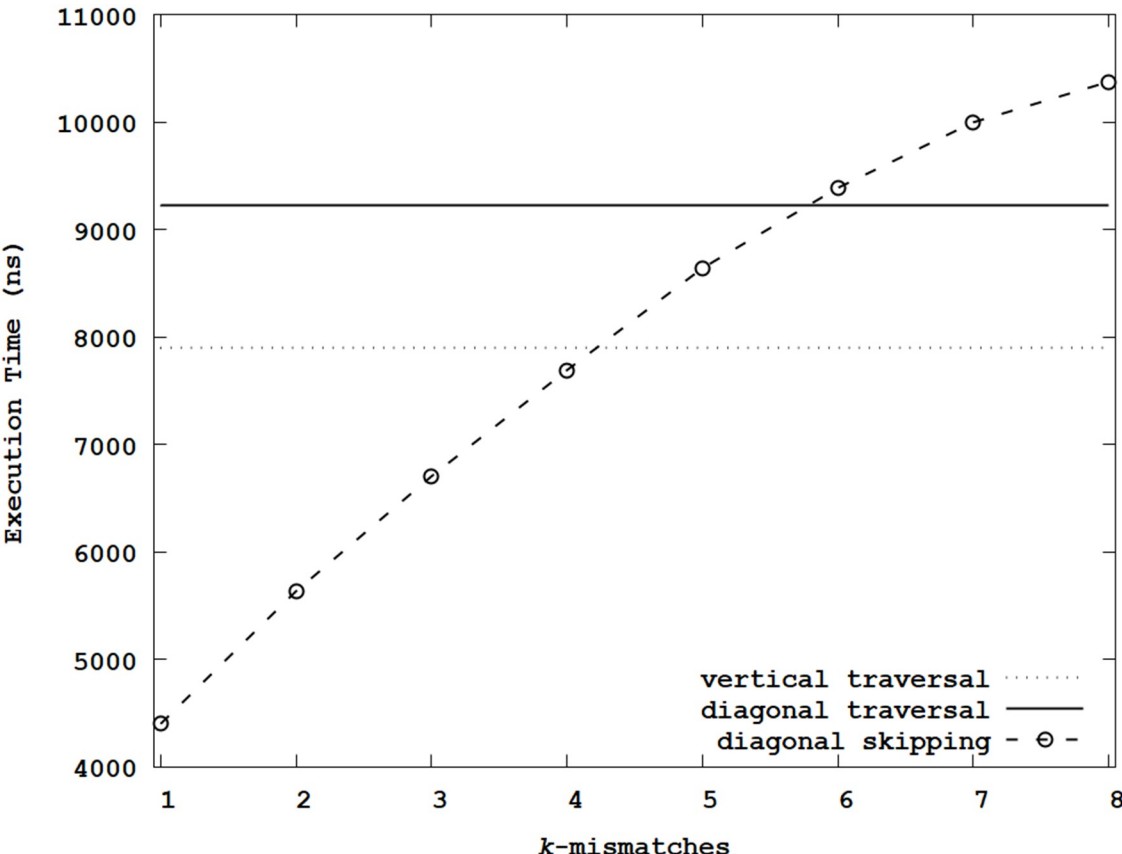

**Fig 6. Average execution times of the Levenshtein distance metric.**

was expected that the execution time can be further reduced because the overhead ratio of conditional statements and reordered memory accesses was smaller.

Fig 8 summarizes the average execution times based on the distance metric considering keyboard character positions. Like Figs 6 and 7, the execution times were evaluated by sweeping *k*. *Diagonal skipping(II)* adopted the skipping method using the pruning register and reduced unnecessary operator evaluations after accessing data-dependent previous steps over *k*. On the other hand, *Diagonal skipping(I)* just used the skipping method using the pruning register. By avoiding unnecessary operator evaluations over *k*, the *Diagonal skipping(II)* can further reduce the execution time when *k* was small. As *k* increased, the difference of the average execution time between *Diagonal skipping(I)* and *Diagonal skipping(II)* was reduced because the number of reduced operator evaluations using *Diagonal skipping(II)* diminished. When *k* = 5, the difference in the average execution times was negligible. When *k* > 5, the average execution time of *Diagonal skipping(II)* was longer than that of *Diagonal skipping(I)*. Besides, when *k* = 8, the average execution time of *Diagonal skipping(II)* was very close to those of the vertical and diagonal traversals. Like the Levenshtein distance and the generalized distance using similarity in shapes, many step calculations can be skipped for small *k*, and the number of skipped step calculations decreased with *k*. However, even when *k* = 8, the proposed method's average execution time was shorter than those of the vertical and diagonal traversals. In agreement with Table 1, the operators' computational costs used in this distance metric can

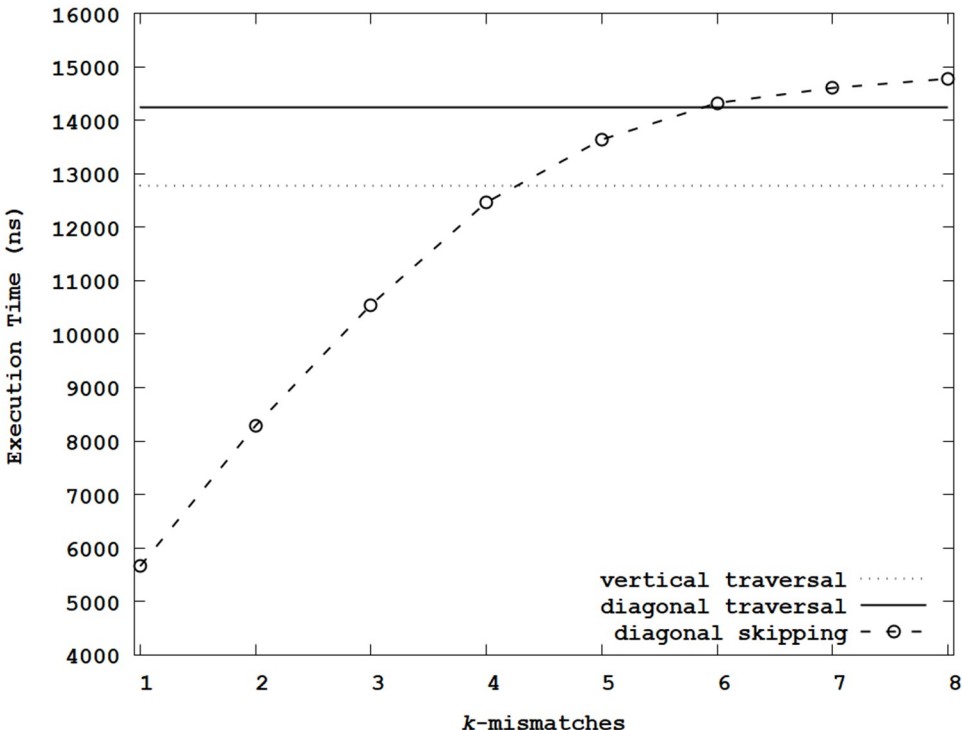

**Fig 7. Average execution times of the generalized edit distance metric using similarity in shapes.**

be high compared with the conditional statements. Notably, compared with the vertical traversal, when $k = 1$, *Diagonal skipping(I)* and *Diagonal skipping(II)* reduced the average execution time by 60.1% and 87.3%, respectively.

Fig 9 illustrates the ratios of the skipped edit distance calculations. This experiment counted two types of skipped edit distance calculations when obtaining $D[i][j]$. When the pruning bit of $(\beta-1)$-th column was '1', the insertion operator was skipped, and only $D[\alpha-1][\beta]$ and $D[\alpha-1][\beta-1]$ were accessed to calculate $D[\alpha][\beta]$. Secondly, if pruning bit of the $\beta$-th column was '1', the next steps in a traversal were skipped because they were over $k$. When $k = 1$, $84.6 \sim 70.9\%$ step calculations were skipped. As increasing $k$, the ratios decreased rapidly, where the decreasing ratios can be different depending on the adopted edit distance metric. When $k = 8$, only $11.0 \sim 3.7\%$ step calculations can be skipped, where the overhead of conditional statements and reordered memory accesses increased the average execution time compared with the vertical and diagonal traversals.

The statistical analysis was performed to know the functional relationship between the execution time and input parameters. As shown in [36], the regression approach was adopted, and the input sequence and pattern lengths were used as input parameters. This evaluation was performed with $k = 2$ for all adopted edit distance metrics. In these regression analyses, the coefficients of determination ($R^2$) can be used to show how much the regression model was fit for the target data [37]. When using the Levenshtein distance metric, $R^2$ was just 0.267. On the other hand, $R^2$s of the generalized edit distance metrics using similarity in shapes (denoted as Shape) and keyboard position (denoted as Keyboard) were 0.575 and 0.818, respectively. These results showed that except for the input sequence and pattern lengths, other overheads could significantly affect the the Levenshtein distance metric's execution time.

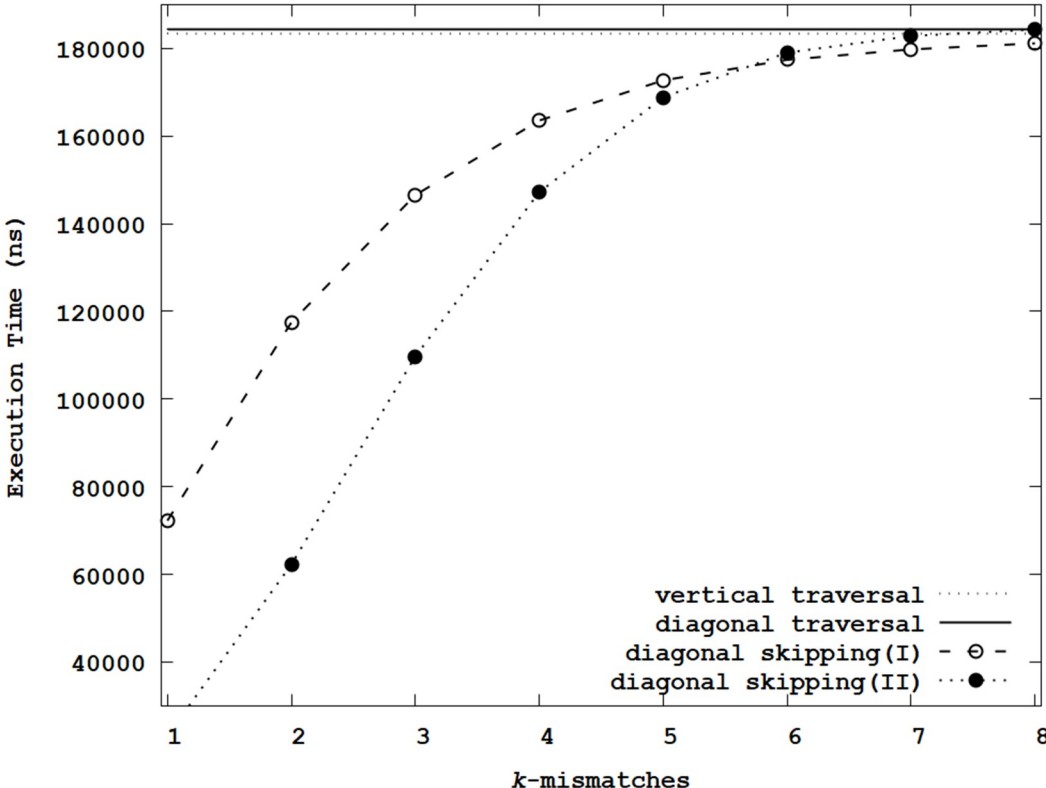

**Fig 8. Average execution times for the generalized edit distance metric using keyboard character positions.**

Table 2 lists the results of the regression analysis for the adopted three edit distance metrics, where *Coef.*, *SE Coef.*, *T*, and *P* denote the coefficient, standard error coefficient, *t*-value, and *p*-value, respectively. Because the *p*-values were small, the input sequence and pattern lengths can be statistically significant. Large *t*-values in Table 2 show that even though the input sequence and pattern lengths were the same, the execution time can be different severely depending on the input sequence and pattern values. Moreover, the coefficient for the pattern lengths was more significant than that of input sequences, which means that the pattern lengths were more critical in the execution time.

## Conclusion

This paper proposes *k*-mismatch approximate string matching for the generalized edit distance. When the generalized edit distance is involved, this paper shows that the step calculations' skipping can reduce the execution time. The proposed method adopts the pruning register to skip step calculations in the diagonal traversals. This paper introduces practical generalized edit distance metrics for the sophisticated experimental environments. The Levenshtein and two generalized edit distance metrics based on similarity in shapes and keyboard character positions are applied to know the effectiveness of the proposed method. In experiments, even though the overhead of conditional statements and reordered data accesses exists in the generalized edit distance metrics, the proposed method can reduce the execution time of *k*-mismatch string matching. Considering the experimental results with realistic edit

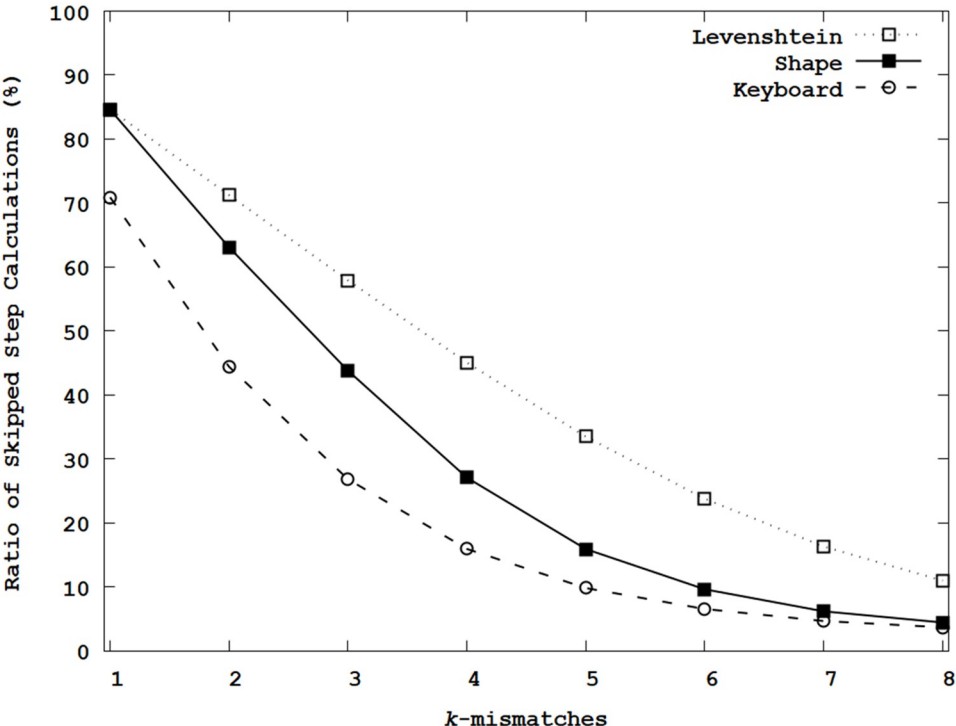

**Fig 9. Ratios of skipped edit distance calculations.**

**Table 2. Statistical analysis using a regression approach according to the edit distance metric.**

| Metric | Term | Coef. | SE Coef. | *T* | *P* |
|---|---|---|---|---|---|
| Levenshtein | Constant | 2328.18 | 22.81 | 102.06 | 0 |
| | length(input) | 34.03 | 1.67 | 20.40 | 2.5E-92 |
| | length(pattern) | 316.26 | 1.67 | 189.85 | 0 |
| Shape | Constant | 96.89 | 29.98 | 3.23 | 1.23E-3 |
| | length(input) | 64.96 | 2.19 | 29.17 | 3E-186 |
| | length(pattern) | 803.49 | 2.19 | 366.95 | 0 |
| Keyboard | Constant | -52085.6 | 329.11 | -158.26 | 0 |
| | length(input) | 1982.56 | 24.07 | 82.37 | 0 |
| | length(pattern) | 15981.31 | 24.03 | 664.92 | 0 |

distance metrics, the proposed skipping method helps reduce the execution time in *k*-mismatch approximate string matching.

## Author Contributions

**Conceptualization:** HyunJin Kim.

**Data curation:** HyunJin Kim.

**Formal analysis:** HyunJin Kim.

**Funding acquisition:** HyunJin Kim.

**Investigation:** HyunJin Kim.

**Methodology:** HyunJin Kim.

**Project administration:** HyunJin Kim.

**Resources:** HyunJin Kim.

**Software:** HyunJin Kim.

**Supervision:** HyunJin Kim.

**Validation:** HyunJin Kim.

**Visualization:** HyunJin Kim.

**Writing – original draft:** HyunJin Kim.

**Writing – review & editing:** HyunJin Kim.

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
