## [Decision Letter · Decision Letter 0]

15 Sep 2020

PONE-D-20-11239

A k-Mismatch String Matching for Generalized Edit Distance using Diagonal Skipping Method

PLOS ONE

Dear Dr. Kim,

Thank you for submitting your manuscript to PLOS ONE. After careful consideration, we feel that it has merit but does not fully meet PLOS ONE’s publication criteria as it currently stands. Therefore, we invite you to submit a revised version of the manuscript that addresses the all the points raised during the review process.

We look forward to receiving your revised manuscript.

Kind regards,

Hans A Kestler

Academic Editor

PLOS ONE

Journal Requirements:

2.We suggest you thoroughly copyedit your manuscript for language usage, spelling, and grammar. If you do not know anyone who can help you do this, you may wish to consider employing a professional scientific editing service.  

Reviewers' comments:

Reviewer's Responses to Questions

**Comments to the Author**

1. Is the manuscript technically sound, and do the data support the conclusions?

Reviewer #1: Partly

Reviewer #2: Yes

2. Has the statistical analysis been performed appropriately and rigorously? 

Reviewer #1: No

Reviewer #2: Yes

3. Have the authors made all data underlying the findings in their manuscript fully available?

Reviewer #1: No

Reviewer #2: Yes

4. Is the manuscript presented in an intelligible fashion and written in standard English?

Reviewer #1: No

Reviewer #2: No

5. Review Comments to the Author

Reviewer #1: The paper covers an interesting technical topic and provides an algorithm within the scope of the topic.

In its current state, I see some flaws of the paper.

- While most of the paper ist written in understandable English, it is clear through the whole paper that the writer is not a native English speaker. I recommend having the paper checked by a native speaker or someone with experience in writing English mauscripts.

- The description of the algorithm is hard to read, mainly because some terms are used before they are explained. Especially the term "arrow" is not defined explicitly. Also, the pseudocode states an implementation step, but the reader has to search for a more detailed description of this initialization.

- In the description of the distance D (beginning in line 75), the difference between x_i and X_i is unclear. Is X_i a string or a character? If it's a string, why is it indexed? If it's a character, why is it written with a capital letter (in contrast to lowercase letters in line 75).

- It appears the algorithm and the datasets used for the analysis are not publicly available.

- The analysis part misses a statistical analysis. This can lead to a misinterpretation of the data: Shorter average running times do not necessarily imply a more efficient algorithm. Also, comparing running times per se can lead to systematic errors in the analyis. E.g. one of the algorithms might have been implemented more efficiently, the computer hardware might be more beneficial for one of the compared programs (e.g. due to cache structure, compiler optimization etc.).

To sum up, I think that the paper needs some work but has potential.

Reviewer #2: The author scrupulously details the approach for reducing the execution time when calculating the generalized edit distance, considering, for example, similar shape characters or keyboard character positions. Through the accurate experimental analysis and the corresponding figures we can visualize how, by means of the proposed method, the execution time is reduced when complicate distance metrics are used. Furthermore the author introduces the disadvantage of using the skipping method when simple metrics are employed.

However some essential introductory notions are ambiguous and need to be better explained.

For example, at line 75, according to the used notation, the input strings X and Y appear to have length i and j respectively, but afterwards, starting from line 76, i and j are used as indices. Even more confusion can be created by considering line 87, where the input sequence X and pattern Y are denoted with subscripts i and j respectively.

Besides certain technical details should be expanded and clarified to ensure better understanding. For instance in Equation 1 the functions of substitution, deletion and insertion should be precisely defined.

6. PLOS authors have the option to publish the peer review history of their article (what does this mean?). If published, this will include your full peer review and any attached files.

Reviewer #1: No

Reviewer #2: No

---

## [Author Response · Author response to Decision Letter 0]

1 Dec 2020

Dear Reviewers,

Thank you for offering the opportunity to revise the paper PONE-D-20-11239 titled: "A k-Mismatch String Matching for Generalized Edit Distance using Diagonal Skipping Method," written by HyunJin Kim to be considered for publication as a research article in PLOS ONE.

Considering reviewer’s comments and concerns, the paper has been revised with careful study in aspects of language, terminology, conveyed meaning, paper format, and grammar. In addition, the revised paper has addressed academic editor and all reviewers’ comments sincerely. There have been several significant modifications as follows: firstly, the terminology and confusing explanation of the edit distance have been revised. Secondly, several terms related to the proposed algorithm has been additionally explained. Finally, we have changed the measurement of the execution time in the proposed method, where the obtained experimental data have shown that the proposed k-mismatch string matching can reduce the execution time using diagonal skipping method in all three different edit distance metrics. We have inserted additional experimental analysis of the ratio of the skipped steps and statistical analysis using a regression approach. 

We believe that this revised draft has solved reviewer’s concerns very well. The detail rebuttal has been uploaded as separate file.

---

## [Decision Letter · Decision Letter 1]

11 Feb 2021

PONE-D-20-11239R1

A k-Mismatch String Matching for Generalized Edit Distance using Diagonal Skipping Method

PLOS ONE

Dear Dr. Kim,

Thank you for submitting your manuscript to PLOS ONE. After careful consideration, we feel that it has merit but does not fully meet PLOS ONE’s publication criteria as it currently stands. Therefore, we invite you to submit a revised version of the manuscript that addresses the points raised during the review process.

Please perform the remaining minor changes raised by the reviewer.

We look forward to receiving your revised manuscript.

Kind regards,

Hans A Kestler

Academic Editor

PLOS ONE

Reviewers' comments:

Reviewer's Responses to Questions

**Comments to the Author**

1. If the authors have adequately addressed your comments raised in a previous round of review and you feel that this manuscript is now acceptable for publication, you may indicate that here to bypass the “Comments to the Author” section, enter your conflict of interest statement in the “Confidential to Editor” section, and submit your "Accept" recommendation.

Reviewer #1: All comments have been addressed

Reviewer #2: (No Response)

2. Is the manuscript technically sound, and do the data support the conclusions?

Reviewer #1: Yes

Reviewer #2: Yes

3. Has the statistical analysis been performed appropriately and rigorously? 

Reviewer #1: Yes

Reviewer #2: Yes

4. Have the authors made all data underlying the findings in their manuscript fully available?

Reviewer #1: Yes

Reviewer #2: Yes

5. Is the manuscript presented in an intelligible fashion and written in standard English?

Reviewer #1: Yes

Reviewer #2: No

6. Review Comments to the Author

Reviewer #1: In the current version of the paper, my previous comments to the last version have been adequately addressed.

Reviewer #2: Numerous of the apported adjustments improved the readability and informativeness of the paper, nevertheless a careful review is necessary for a complete perception of the study.

- A further accurate copy edit for language usage and grammar is mandatory. In fact a frequent mistake, is the usage of the verb in the third person singular when not required, for example at lines 98, 410, 412, but may not be the only ones. Other repetitive misspellings are the quoted comma (for example line 257: '1,') and capital letter after comma (see line 16). There are also lexical inaccuracies as close statement repetitions (for instance 343-345 and 349-350) or unclear references (at 346-348 one does not promptly associate "these experiments" with "the diagonal skipping method").

- The distance (as defined in lines 76-77) is the minimum number of operations to convert the input string into the pattern, it follows that the operations are always applied on the input string and never on the pattern. Therefore in expression (1) should be correct D(X_α-1, Y_β) + insertion(y_β) and D(X_α, Y_β-1) + deletion(x_α). This statement is also confirmed by the explanation of the example in Fig 1 (b), lines 112-118. Moreover the example in lines 104-109 needs a better adapted description.

- It is suggested to revisit the notations used to represent strings, substrings, the respective lengths and the indices to identify any character in the string/substring. There are misconceptions when the subscript refers to the length of a string/substring and when it refers to any character of the string/substring. For instance at line 267 the content suggests that D[α][1] wants to indicate the whole second column in the two-dimensional array D, however 'α' defines the length of the input substring. Moreover according to the employed notation D[α][β] (line 269) identifies only the rightmost bottom cell of D, but here it is intended to identify any cell of D.

- Comparing the definition of ‘traversal’ specified in 120-121, its usage in the pseudocode and in lines 229-230, the difference between arrow and traversal is unsettled.

7. PLOS authors have the option to publish the peer review history of their article (what does this mean?). If published, this will include your full peer review and any attached files.

Reviewer #1: No

Reviewer #2: No

---

## [Author Response · Author response to Decision Letter 1]

18 Mar 2021

Dear Reviewer,

Thank you for reviewing this paper PONE-D-20-11239R2 titles as "A k-Mismatch String Matching for Generalized Edit Distance using Diagonal Skipping Method." 

Considering reviewers' comments, this paper has been revised with careful study in aspects of paper format, language, terminology, conveyed meaning, and grammar. 

Several typos and missing information have been corrected. In addition, the revised paper addressed all reviewers' comments. The detailed response is uploaded as PDF file.

---

## [Editor Report · Decision Letter 2]

20 Apr 2021

A k-Mismatch String Matching for Generalized Edit Distance using Diagonal Skipping Method

PONE-D-20-11239R2

Dear Dr. Kim,

We’re pleased to inform you that your manuscript has been judged scientifically suitable for publication and will be formally accepted for publication once it meets all outstanding technical requirements.

Kind regards,

Hans A Kestler

Academic Editor

PLOS ONE
---

## [Editor Report · Acceptance letter]

23 Apr 2021

PONE-D-20-11239R2 

A *k*-Mismatch String Matching for Generalized Edit Distance using Diagonal Skipping Method 

Dear Dr. Kim:

I'm pleased to inform you that your manuscript has been deemed suitable for publication in PLOS ONE. Congratulations! Your manuscript is now with our production department. 

Kind regards, 

on behalf of

Prof. Hans A Kestler 

Academic Editor

PLOS ONE